# Lead Leakage of Pb-Based Perovskite Solar Cells

**Kai Ma** [1], **Xiaofang Li** [2,*] , **Feng Yang** [2] **and Hairui Liu** [3,*]

1 School of Chemistry and Chemical Engineering, Henan Normal University, Xinxiang 453007, China
2 Henan Key Laboratory of Photovoltaic Materials, School of Physics, Henan Normal University, Xinxiang 453007, China
3 College of Material Science and Engineering, Henan Normal University, Xinxiang 453007, China
* Correspondence: lixiaofang@htu.edu.cn (X.L.); liuhairui1@126.com (H.L.)

**Abstract:** As a novel technology, perovskite solar cells (PSCs) have attracted worldwide attention due to their high photoelectric conversion efficiency (PCE) and low fabricating cost. Moreover, with the development of this technology, PSCs have achieved a great breakthrough in PCE. However, the heavy metal element Pb in PSCs does harm to human health and ecological environments, which restricts the further application of Pb-based PSCs. Under certain circumstances, the leakage of lead will cause serious pollution to the environment. The purpose of this review is to summary and discuss the way of lead leakage suppression. Among them, we pay more attention to the method of packaging technology, chemisorption procession and the limitations of each method. Finally, strategies of highly PCE and non-toxic perovskite devices are proposed.

**Keywords:** lead leakage; encapsulation; chemical adsorption; lead-free perovskite; lead contamination

## 1. Introduction

Perovskite solar cells (PSCs) have attracted wide attention from scientists due to their excellent photoelectric characteristics, such as long carrier length, high carrier mobility, high absorption coefficient, and low trap density. In terms of manufacturing costs, perovskite solar cells have low prices because of the low cost of rich materials (Ti, Pb, I, Cl, Br, etc.), making them the most competitive technology in the solar cell industry [1–3]. The power conversion efficiency (PCE) of single-junction perovskite solar cells has exceeded 25%, surpassing the record set by copper indium gallium selenium (CIGS) solar cells and approaching that of the best crystalline silicon solar cells [4–6]. Furthermore, the PCE of perovskite/silicon tandem solar cells have exceeded 29% [7–11]. Increasing teams are devoted to the research of PSCs, which undoubtedly accelerates the further commercial development of PSCs.

Lead element is often involved in PSCs thanks to their high crystal symmetry, unique atomic electron fusion, and strong spin orbital coupling capacity. In general, the common device structure containing Pb is $APbX_3$, where A is an organic or inorganic cation (such as methylammonium, formamidine, or $Cs^+$) and X is a halide ion (mainly $I^-$ or $Br^-$) [12–14]. The results show that the Pb-based perovskite possessed the highest photoelectric conversion efficiency in all the perovskite solar cells [13,15]. Although the photovoltaic properties of Pb-based perovskite solar modules are excellent, the safety hazard brought by them cannot be ignored at the same time. It is worth noting that Pb-based perovskite contains much water-soluble lead salts as degradation products [16]. Even a small amount of Pb leaking into the environment is toxic to human health and the environment. Lead or its compounds can be absorbed by the human body through breathing, eating, skin absorption, and the like. According to the relevant studies, 20% to 80% of the ingested lead can be absorbed by a human body, and children have a higher absorption capacity of lead. Excessive lead intake can inhibit the normal synthesis of proteins and cause a number of healthy problems. For example, the increased levels of lead in the blood have a harmful

effect on infants' and children's behavior, cognitive performance, pubertal development and hearing ability [17]. As for adults, lead can cause a series of cardiovascular, central nervous system, kidney and fertility problems. Worse still, lead can also stunt early fetal growth during pregnancy. In the meantime, lead also does harm to the environment, and can inhibit the growth of growing plants [17–21]. The main toxic substance released by perovskite solar cells when they are decomposed is lead iodide. To illustrate this point, Kwak et al. studied the toxicity of $PbI_2$ to the embryos of two species of fish (zebrafish and medaka) and focused on the chemical speciation of $PbI_2$ in the culture medium of embryos to characterize the toxicity of lead iodide to organisms. Research results reveal that the mortality, deformity, hatching failure, growth inhibition, and other pathological changes are increased in fish which are exposed to lead iodide [11,22]. Hasan Ul Banna et al. conducted a test of lead exposure on mice and found that Pb element can induce anxious behavior and memory learning disorders in mice. The memory learning disability of mice exposed to Pb could be improved by training, but that of mice exposed to Pb environment from the embryonic stage was always lower than the initial level of mice who without contact to Pb (shown in Figure 1). Mice exposed to lead as fetuses suffered more severe neurobehavioral changes and liver damage [23]. Therefore, in order to realize the commercial application of perovskite solar cells in a large area, we must first focus on solving the problem of lead toxicity [24].

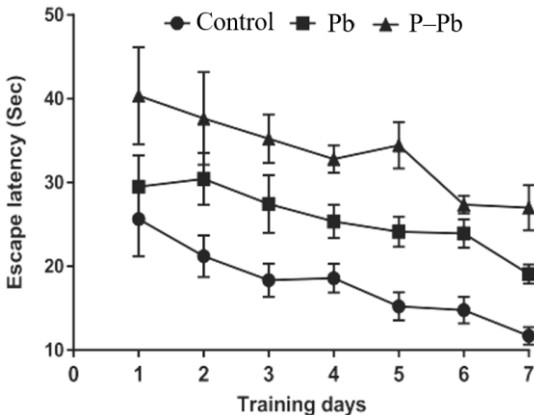

**Figure 1.** Effects of Pb on learning and spatial memory in mice. Mice in the Control group were not exposed to lead, mice in the Pb group were exposed to lead, and mice in the P–Pb group were exposed to lead during pregnancy and after birth. Sec represents the learning and spatial memory ability of mice, and the lower the Sec value is the stronger learning and spatial memory ability of mice. Reproduced with permission from [23]. Copyright 2021, Springer Nature.

Normally, the lead in perovskite crystals is contained in the PV module and does not leak into the environment. However, in extreme cases, such as hail, earthquakes, and extreme temperatures, PV modules can be damaged. Perovskite materials are affected by moisture and oxygen, and lead-based perovskites tend to release toxic lead iodide as degradation products that can be transmitted from soil to homes and/or drinking water [25,26]. Scientists have investigated the strategies to prevent lead leakage, holding that the ideal prevention strategy is to trap lead ions quickly. For example, perovskite can run for a long time when broken down by rain. In addition to protecting against outside factors such as rain, dust, or UV rays, there are many other effective ways to prevent Pb leaks. The encapsulation method can greatly increase the perovskite module's ability to withstand the external impact force and protect the device module from damage. Chemisorption can realize the adsorption of $Pb^{2+}$ in the process of $Pb^{2+}$ leakage, and fix $Pb^{2+}$ in the component to avoid the release of lead into the atmosphere. Nonetheless, these methods cannot fundamentally reduce the concentration of lead ions released into the environment, so the study of lead-free perovskite can approach the environmental pollution of lead ions from the root. In the following part of this paper, we mainly introduce

these three methods to solve the lead leakage issue and put forward the prospect of further commercial application after solving the lead leakage problem.

## 2. Mechanism of Lead Leakage from Pb-Based Perovskite

Poor stability of the Pb-based perovskite solar cells is the main cause of the lead leakage. There are two factors affecting the stability of PSCs: internal factors and external factors. Internal instability factors include perovskite structure, transport layer materials, defects caused by ion migration, and degradation of perovskite materials. It can be improved by regulating the crystal structure of perovskite, controlling ion migration, passivation defects, and adopting relatively stable transport layer materials. External factors such as moisture and oxygen generally addressed through encapsulation technology. For example, HI generated by hydrolysis of $CH_3NH_3PbI_3$ ($MAPbI_3$) would decompose into $I_2$ under ultraviolet or light irradiation, which further promotes the degradation process [27]. The decomposed $Pb^{2+}$ exists in the form of $PbI_2$, which easily leaked into the environment and caused environmental pollution [28]. Hailegnaw et al. proved that lead loss rate was as high as 72% after 5 min of rain [29].

$$CH_3NH_3PbI_3 \xleftrightarrow[hv]{H_2O} CH_3NH_3I + PbI_2 \tag{1}$$

$$CH_3NH_3I \leftrightarrow CH_3NH_2 + HI \tag{2}$$

$$2HI \xrightarrow{UV} H_2 + I_2 \tag{3}$$

$$4HI + O_2 \leftrightarrow 2I_2 + 2H_2O \tag{4}$$

Compared to MA, FA is a larger molecule with a smaller dipole moment. For stronger binding between FA and halides, FA-based perovskite has higher stability attributed by less halide ion migration [30]. However, when the temperature exceeds 230 °C, such as at the scene of a fire, decomposition still occurs, and the decomposition reaction is shown by the equation below:

$$FAPbI_3 \leftrightarrow PbI_2 + FAI \tag{5}$$

It can be seen that $PbI_2$ is the main decomposition product of $MAPbI_3$ and $FAPbI_3$ under hot and humid conditions [31]. Normally, the Pb-based perovskite solar cell is safe even if the PSCs degrades. However, in extreme conditions such as wind, snow, hail, fire etc., the PV module may be subjected to strong mechanical shocks, which would result in leak into the environment. Therefore, it is extremely important to find the ideal packaging material.

## 3. Encapsulation

The degradation of perovskite solar cells is mainly caused by oxygen and water [32,33]. Encapsulation is a common protection method that can inhibit moisture from entering equipment, reduce the oxygen and water penetration to an almost negligible level, and curb the outflow of decomposed lead elements. As a qualified packaging layer, it should have a high performance of shielding oxygen and moisture, and can work well under extreme weather to improve the life of a device [34–36]. At the same time, it should also have great thermal stability and good light transmittance. A few typical packaging methods to suppress lead leakage are given below.

Ethylene vinyl acetate (EVA), as a copolymer of ethylene and ethylene acetate, is a common solar cell packaging material (covering layer, pooping agent and substrate). Moreover, EVA also features desirable light transmittance and elasticity, adhesion strength with glass, melt fluidity, and other advantages. Accordingly, EVA is widely used as a packaging material, with nearly 80% of photovoltaic (PV) modules utilizing EVA as an encapsulation material. Common EVA encapsulation method is shown in Figure 2a. Crosslinked EVA wafer is characterized by favorable transparency, and its transmittance in 400~1100 nm is about 92.8%, which can help realize the protection of components without

affecting the device efficiency [37,38]. The degree of crosslinking is an important index of EVA encapsulation agent, which represents the degree of crosslinking of a polymer chain. If the crosslinking degree is too high, EVA will become brittle and will not withstand the external impact force. By contrast, if the crosslinking degree is too low, the aging resistance will be reduced, and it cannot meet the requirement of creep resistance. Taken together, the appropriate crosslinking degree is in the range of 75%~90% [39]. Crosslinked EVA tablets have such advantages as low cost and good transmittance. However, as a packaging material, an EVA sheet still faces a serious problem: EVA sheet will be aged and degraded in a long-term ultraviolet and hot atmosphere (shown in Figure 2b) making the color of the polymer film change from transparent and colorless to yellow or brown [40]. This color change will greatly reduce the light absorption range and the efficiency of solar cells. Researchers have found that the performance of EVA can be improved by adding antioxidants, ultraviolet absorbers, and light stabilizers, but this means cost increase, robbing EVA sheet of the advantage of low cost. Moreover, when the temperature of the EVA-encapsulated battery over 120 °C, the perovskite material turns yellow, indicating that the perovskite layer is decomposed. This decomposition mechanism is attributed to acetic acid in EVA [21,41]. To reduce the affection of acetic acid in EVA on the perovskite layers, M. D. McGehee further replaced EVA with polyolefin (POE) [42]. However, it was found that both the perovskite absorber and the charge transport layer degraded at high temperatures, which leads to the photovoltaic characteristics of the cells significantly reducing at 140–160 °C. Therefore, the traditional POE and EVA seem not the idea package materials for perovskite solar cells. Compared with these two packaging materials, PU (polyurethane) shows its significant advantages. The PU packaging is carried out at a relatively mild temperature of 80 °C, at which temperature the perovskite absorption layer would not decompose. Moreover, the perovskite absorption layer packaging with PU was not destroyed even at 120 °C. Zhengyang Fu proved that there is no performance degradation observed after keeping the PU-encapsulated battery at 85 °C for more than 325 h [43]. In summary, PU can be used as a simple and effective way to package perovskite solar cell module.

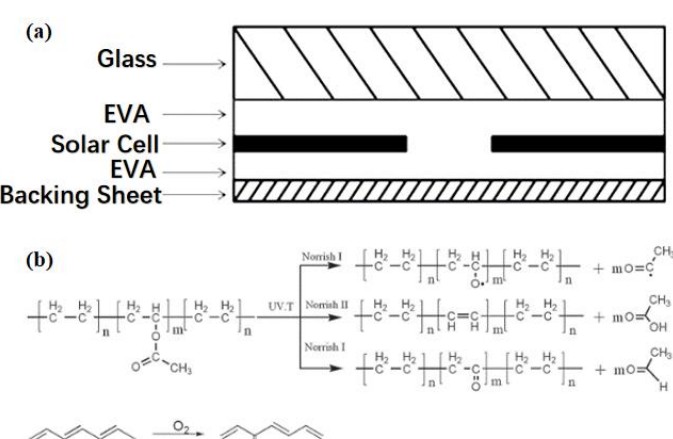

**Figure 2.** (**a**) Structure of EVA encapsulated solar cells. (**b**) Aging degradation mechanism of EVA. Reproduced with permission from [39]. Copyright 2015, John Wiley and Sons.

In addition, Kim et al. found that there is no $PbI_2$ generated after exposed $MAPbI_3$ based perovskite solar cell to the air for 26 days when the Au electrodes of the device were attached by Poly(p-chloro-xylylene) [44]. Further study proved that poly(p-chloro-xylylene) encapsulation can effectively prevent water into the PSCs module, and then inhibit the decomposition of perovskite. In the same way, Lei Shi et al. encapsulate PSCs device by polyolefin on the top of the electrode. Then, the device was covered by a piece of glass. The experiment results indicate that the encapsulated equipment has no Pb precipitation for more than 1800 h in the humid heat environment. This method not only prevents the entry

of moisture, but also inhibits the output of decomposing gases such as $CH_3I$ or $NH_3$ [45]. Moreover, Jie Yin et al. used UV glue to wrap PSCs. The UV-vis absorption spectrum results showed that the Pb-based perovskite solar cells packaged by UV glue could remain their photoelectric properties over 5 years [21,41,46].

Thermosetting epoxy resin has high mechanical strength, excellent dimensional stability, and good chemical resistance, so it is widely used in the packaging of photovoltaic devices [47]. For example, Yan Jiang et al. [48] found that the key to reducing lead leakage was the self-healing ability of certain polymers when heated above their glass transition temperature and inferred that an increase in the polymer's self-healing ability would significantly reduce lead leakage. Epoxy resin (ER) is another effective encapsulation material. The team demonstrated that ER's self-healing properties and improved mechanical strength could effectively prevent lead leakage, and lead leakage from lead halide perovskite PV products can be reduced by a factor of 375 if properly packaged. The ER film can be formed by a mixture of diglycidyl ether bisphenol A (DGEBA), n-octylamine (OA), and diphenyl phenylenediamine (MXDA) diglyceride ether. Jiang Yan et al. studied self-healing ER (DGEBA: OA: MXDA = 4:2:1) membranes by physical cross-linking of DGEBA and OA and chemical cross-linking of MXDA and DGEBA. In a contrast test, the test subjects were divided into four groups: A, B, C, and D. The encapsulation method of the test was as followed: (A) No package, (B) top package with perovskite solar modules/UV resin/glass, (C) top and bottom package with glass/Surlyn resin/perovskite solar modules/UV resin/glass, and (D) top and bottom package with glass/epoxy resin/perovskite solar modules/UV resin/glass. By simulating the amount of lead leakage under extreme weather conditions, such as acid rain and hail, it proved that an ER film had a self-healing characteristic and could realize self-healing using the heat provided by the sun. Therefore, self-healing could be achieved during the operation of solar cells, thereby greatly reducing the amount of lead leakage. By simulating the extreme weather test data, it proved that the lead leakage rate of the perovskite solar cell (method D) was greatly reduced by ER thin film encapsulation. Compared with the encapsulation method using glass cover at the module edge (method B), the lead leakage rate of Pb could be decreased from 30 to 0.08 mg h$^{-1}$m$^{-2}$, and the lead leakage rate of the ER encapsulation method was reduced by 375 times (shown in Figure 3) [48]. Through comparison tests under four different weather conditions, the team noted that the lead leakage in group D was much smaller than that in other groups, which could effectively control the total leakage of self-healing polymer packaging under extreme weather conditions in slow response time. The high mechanical strength and self-healing mechanism of ER films can effectively reduce the leakage of lead in perovskite solar cells, which provides a new insight for the packaging method of perovskite solar cells. By applying self-healing and lead-adsorbed ion gel sealers to the front glass surface and between the electrodes and the packaging adhesive glass, the perovskite module can be physically prevented from seeping water into the perovskite module if the packaging glass is damaged, when chemically trapping lead that may leak [49].

Atomic layer deposition (ALD) technology looks like an ideal packaging method for the perovskite solar cells, which can prepare thin film encapsulation layer (TFE) with multilayer under the condition of adjusting appropriate parameters. The most common approach is depositing encapsulation layer with inorganic ($Al_2O_3$)/organic (pV3D3) alternately structure via ALD onto the top of the PSCs (Figure 4a) [50]. While the inorganic layer blocks the external environment, the organic layer enhances the flexibility of the film and increases the smoothness of the PSC surface (shown in Figure 4c). This special structure can effectively delay the penetration of water and oxygen, greatly improving the barrier performance of the packaging layer, minimizing equipment damage, and consequently improving the long-term stability of the device. However, an ignored problem is that long worktime of ALD at 90 °C will bring irreversible damage to PSCs, because this temperature is too high for depositing perovskite films (Figure 4b,d). Therefore, the research on the deposition of low-temperature ALD or even room temperature ALD is an urgent problem to be solved [51–54].

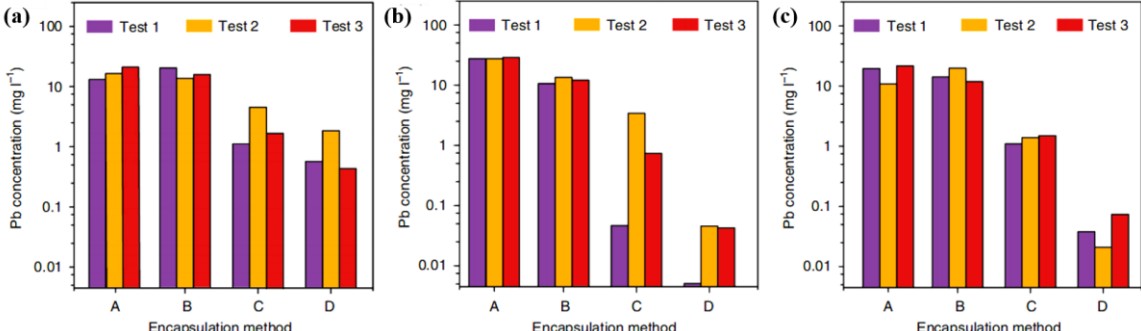

**Figure 3.** The concentration of Pb leaking into the water. Three parallel experiments were carried out for each package method to reduce the error. (**a**) Perform a drip test on the damaged PV module. (**b**) The damaged perovskite solar module was dripped and then kept in simulated sunlight for 4 h (45 °C) before being injected again. (**c**) The damaged perovskite solar modules are heated at 45 °C for 4 h and then dripping with water. No matter what the conditions, group D always has the lowest lead leakage. Reproduced with permission from [48]. Copyright 2019, Springer Nature.

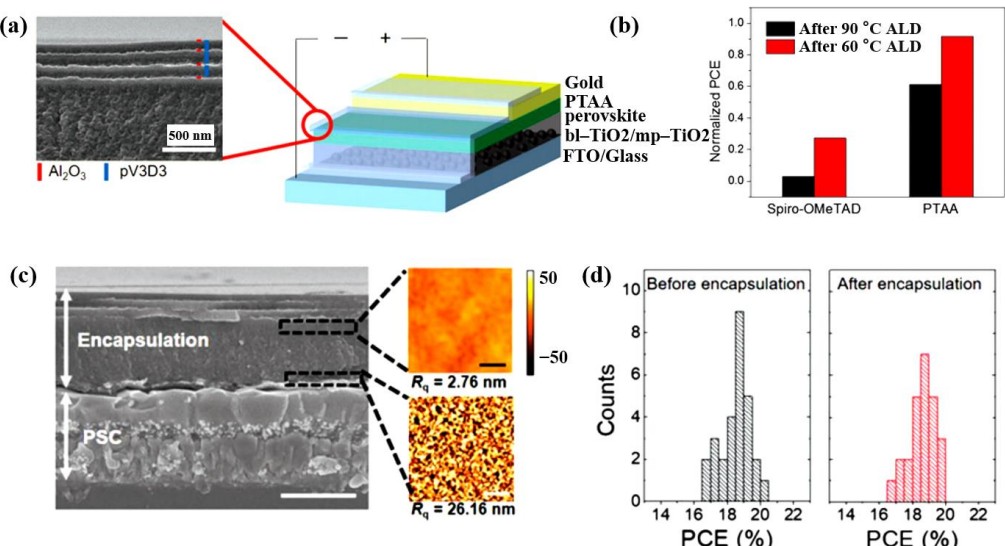

**Figure 4.** Perovskite solar modules packaged by the TFE method. (**a**) SEM image of cross section of TFE layer and schematic diagram of perovskite solar module packaged by TFE. (**b**) Changes of PCE in HTM layer after 60 °C and 90 °C ALD process. (**c**) Cross sectional SEM image of PSC encapsulated by CVD. The surface smoothness is improved. (**d**) PCE distribution of samples before and after encapsulation. Reproduced with permission from [50]. Copyright 2017, John Wiley and Sons.

Flexible perovskites usually need to be encapsulated in organic materials because organic packaging materials can be synthesized through organic molecules with specific properties by changing the energy level, molecular weight and solubility of their own molecules [55]. For instance, Monojit Bag et al. demonstrated that a cross-linked self-healing polymer network of Polyisobutene (PIB) could be used as a raw material for a fully printable flexible solar cell as a self-healing sealant to protect perovskite cell components. These sealants could be applied to flexible perovskite solar modules by drip casting, rotary coating, or blade coating. The team indicated that PIB based devices with cross-linked sealants enjoyed good self-healing property and stability [56]. It can be seen that self-healing polymers can be directly used as packaging materials to achieve good self-healing ability, thus improving the durability of PSCs and preventing lead leakage [57].

## 4. Chemical Adsorption

While packaging methods can greatly reduce lead leakage from devices into the environment, there still be some lead leakage during extreme weather when solar panels can be severely damaged. The use of packaging technology often entails increased costs. Therefore, finding other methods for curbing lead leakage is of urgent necessity. In this case, chemisorption is an economical and effective method, given its high binding energy with lead ions in a mesoporous structure. Cation exchange resin (CER) with a high adsorption capacity and adsorption rate for lead has aroused wide interest from researchers. For illustration, Shangshang Chen et al. further investigated CER by adding sulfonic acid groups to CER materials to make $Pb^{2+}$ adsorbents. Because the sulfonic acid group had a strong bond for $Pb^{2+}$, $Pb^{2+}$ could be firmly adsorbed on the surface of CER to prevent the outflow of $Pb^{2+}$ (shown in Figure 5a). The mesoporous CER layer consists of a number of nanoparticles with an average size of 50 nm that form a dense layer of CER on top of the copper electrode (as shown in Figure 5c–e). This mesoporous structure can significantly increase the surface area to adsorb more lead. The team tested the lead adsorption rate of CER films rinsed with lead-contaminated running water by preparing the CER films on a glass sheet based on ultrasonic-assisted suspension of CER powder in isopropyl alcohol, and then the CER precursor solution was coated on a glass substrate with a thickness ranging from 300 to 1300 nm, it was found that the 300 nm thick CER layer coated on the glass immediately reduced the lead content of the running water by 30% and was independent of the initial lead concentration in the solution. The concentration of $Pb^{2+}$ in water did not decrease significantly with the increase of thickness, indicating that the diffusion rate of $Pb^{2+}$ to the CER surface was limited (Figure 5b). The water contact angle of ~13° was so small that water could easily penetrate into the porous structure of CER, thus promoting the adsorption of $Pb^{2+}$. The PSCs with a CER layer maintained the lead absorption capacity after 600 h of irradiation in the experiment of strong ultraviolet radiation simulation, showing good photostability. In the drip test, lead leakage was reduced by 98% to only 14.3 ppb (Figure 5f,g) [58]. This low lead leakage indicated that CER adsorbed a large amount of $Pb^{2+}$.

Various defects in perovskite films can form deep energy wells that affect carrier transport and device performance. The defect surface has large surface energy and decomposition energy, which can lead to spontaneous relaxation of perovskite surface and leakage of lead ions [59]. As the recombination center, unsaturated Pb can cooperate with a mercaptan group to reduce recombination. Qingrui Wang et al. used 1,2-ethanol instead of mercaptan (1,2-EDT) for surface treatment of perovskite films to form Pb-S bonds. The modification could strengthen the Pb-I bond and passivate the Pb-suspended bond, and the perovskite film delayed the dissolution of lead iodide into the water, thus inhibiting the leakage of Pb. Compared with the conventional film (49.19), the treated perovskite film had a larger water contact angle (71.83), indicating better hydrophobicity and moisture resistance, which could prevent the decomposition of perovskite caused by water entering the devices. The treated perovskite film postponed the dissolution of lead iodide into the water, ultimately reducing lead leakage [60].

The chelating resin features good selectivity, pre-concentration factor, binding energy, and mechanical stability for the removal of heavy ions. It can smoothly carry out the regeneration of multiple adsorption-desorption cycles and enjoys good reproducibility in terms of adsorption characteristics [61]. XAD resins have good physical properties, such as porosity, uniform high surface area, pore size distribution, and chemically uniform non-ionic structure. They are used as a carrier for fixed chelating agents to remove metals. V Tharanitharan et al. prepared a new modified Amber lead XAD-7HP resin using harmless sodium dioctyl succinate (DOSSS) and EDTA-disodium salt (chelating agent). After 7% sodium chloride treatment, the modified XAD-7HP resin could successfully recover lead (II) ions, and the removal rate of Pb (II) could reach 99% at the adsorbent dosage of 0.9 g/100 mL [62].

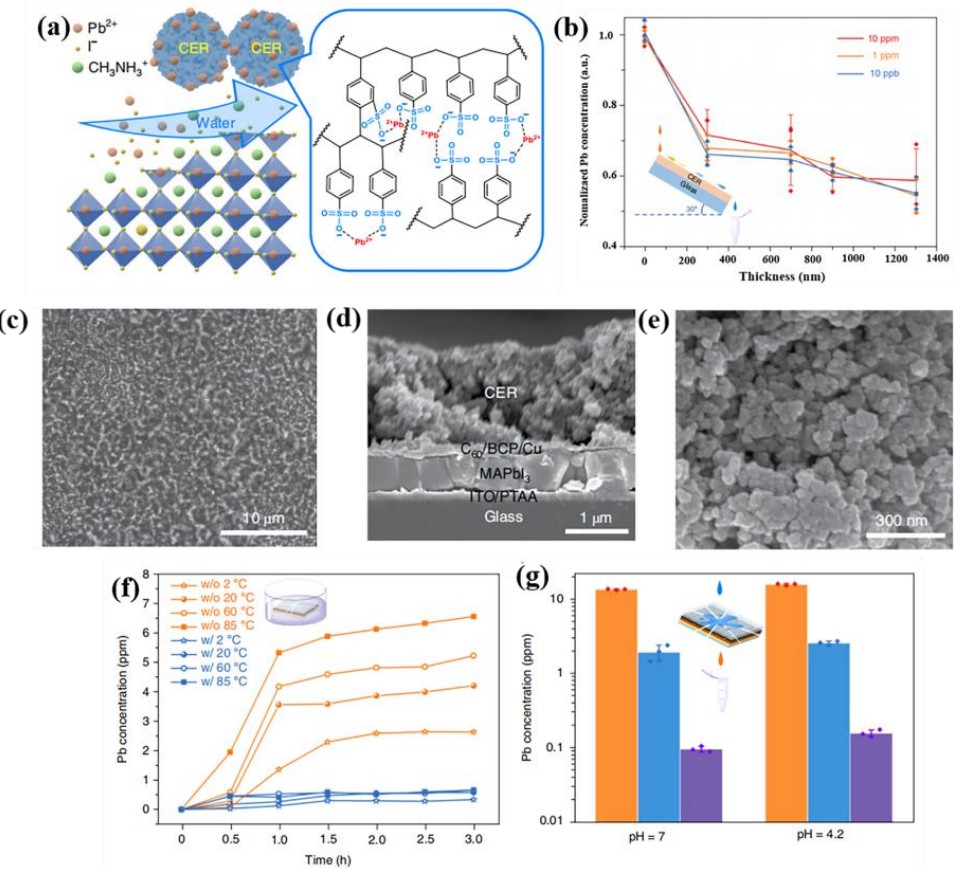

**Figure 5.** (**a**) Schematic diagram of lead leakage prevention mechanism. (**b**) Different concentration of Pb$^{2+}$ aqueous solution at different initial concentrations on glass with CER layers of different thicknesses of 15 cm long. (**c**–**e**) SEM image of top view and cross section of MAPbI$_3$ device with CER-coated and the top of the copper electrode with CER. (**f**,**g**) Results of immersion tests on damaged micromodules with or without CER coating. Reproduced with permission from [58]. Copyright 2020, Springer Nature.

Xun Li et al. deposited a transparent Pb-absorbing P, P′-di(2-ethylhexyl) methanediphosphonic acid (DMDP) film on the glass. DMDP coatings with a thickness ranging from 0.7 to 6.89 µm were highly transparent and had good light transmit ability, which would not have adverse effects on the efficiency of the device [63]. The two phosphate groups in each DMDP molecule could be strongly bound to a Pb$^{2+}$ ion, and when soaked in water, the suction plates on either side expanded to absorb lead rather than dissolve it, thus maintaining structural integrity for easy collection of lead after damage. Li Xun et al. proposed a new method for suppressing lead leakage using a standard solar vinyl acetate (EVA) film and pre-laminated P, P′-di(2-ethylhexyl) methanediphosphonic acid (DMDP) to form a thin layer similar to tape. This layer could be attached to the sides of perovskite solar cells. Whether the solar cell was an n-i-p structure or a p-i-n structure, it could be tightly bonded with the glass surface and became transparent, so it did not affect the transmittance and exerted no adverse effect on the normal photovoltaic performance. The tape could be integrated into the packaging material of the unit at a later stage, so there was no strong dependence on the PSC manufacturing process. The process of making DMDP absorption layer is shown in Figure 6. Our research group's experiments displayed that the presence of lead absorption bands did not reduce the efficiency and stability of the PV module. When PV module was damaged due to extreme weather, the tape absorbed large amounts of leaked lead and maintained SQE above 99.9% for 7 days, demonstrating excellent lead leak suppression properties [64].

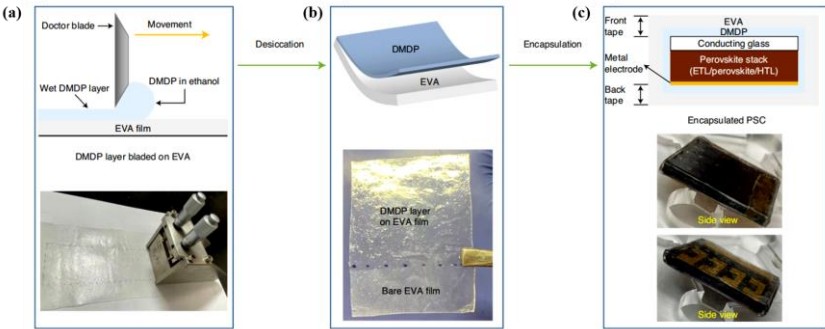

**Figure 6.** A schematic of DMDP applied to EVA film and a photo of PSC made with EVA tape. Reproduced with permission from [64]. Copyright 2021, Springer Nature.

Chen Junjun modified tin dioxide (tin oxide) layer with sodium phosphate ($Na_3PO_4$), which provided a new way to solve the problem of lead leakage (Figure 7a). $PO_4^{3+}$ groups can chelate with dissolved Pb ions to form water-insoluble compounds of $Pb_2PO_4I$, thus reducing Pb leakage. From the XRD patterns in Figure 7b, we found that $SnO_2$-based perovskite film was decomposed into pure $PbI_2$ after soaking in water, but no lead iodide diffraction peak was detected in the XRD patterns of $SnO_2$: $Na_3PO_4$ sample, which revealing that $PO_4^{3+}$ groups could chelate with dissolved $Pb^{2+}$. In addition, when dipped the samples into water, the $Pb^{2+}$ concentration of $Na_3PO_4$ containing film slightly increased to 0.2 ppm in 30 min, while that of the control gradually approached the maximum of 1 ppm in 30 min (Figure 7c). Furthermore, $Na_3PO_4$ containing layer not only improved the performance of the device, but also captured most of the dissolved lead in the water. The inclusion of phosphate facilitated charge transfer and passivated the buried perovskite interface, resulting in a substantial increase in device efficiency of up to 23% with negligible hysteresis. More importantly, the phosphorylated tin oxide layer had a high lead adsorption capacity. Due to the numerous anchor points of oxygen solitary pairs, the isolation efficiency reached 79.6%, which could convert dissolved lead into insoluble compounds in water, the concentration of $Pb^{2+}$ decreased to 2 ppm after 20 min (Figure 7d) [65].

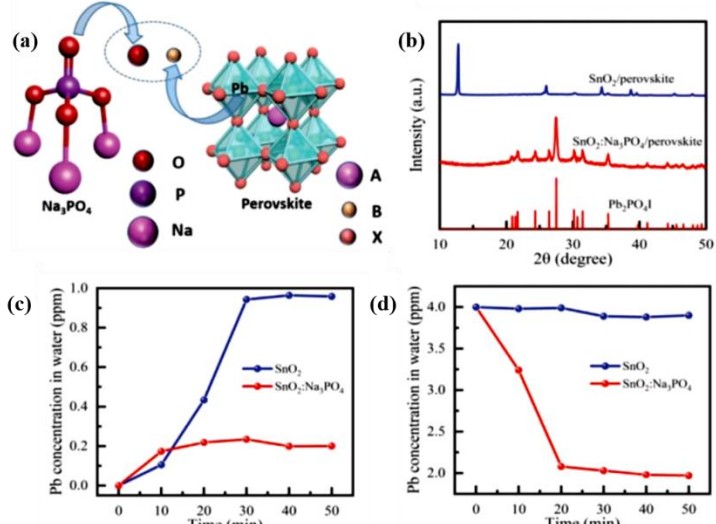

**Figure 7.** (**a**) Schematic diagram of Pb adsorption. (**b**) XRD patterns. (**c**) Transformation of $SnO^{2-}$ and $SnO_2$:$Na_3PO_4$ based perovskites in water. (**d**) Measurement of lead leakage and comparison of lead retention in damaged PSCs with and without $Na_3PO_4$. Reproduced with permission from [65]. Copyright 2022, John Wiley and Sons.

It is worth noting that both encapsulation and chemisorption have advantages and disadvantages (shown in Table 1). The encapsulation layer can effectively inhibit the water

vapor and oxygen into the battery, prevent the leakage of lead, and thus significantly enhance the stability of the device. Compared to the encapsulated technology, chemisorption has a more obvious result. As the perovskite layer decomposes, the chemisorption layer can absorb lead and/or form chelates in the first place to avoid lead leakage. However, when the chemisorption layer absorbs lead in large quantities or forms chelates, the perovskite lattice may collapse due to the reduction of lead, eventually result in the PCE dropped [66]. Therefore, more efforts should be made to design a new chemisorption layer which can suppress the lead leakage and protect the perovskite lattice as well. In addition, the encapsulation layer and the chemisorption layer can be used together, which would realize a dual lock system.

**Table 1.** Advantages and disadvantages of encapsulation and chemisorption.

|  | Encapsulation | Chemical Adsorption |
|---|---|---|
| Cost | High | Low |
| Whether mechanical strength can be improved | √ | × |
| Whether the amount of lead leaking into the environment can be reduced | × | √ |

## 5. Lead-Free Perovskite Solar Cells

Although the above methods can effectively reduce lead leakage, there exists a small risk of lead leakage. In order to completely avoid the risk of lead leakage contaminating the environment, the most effective method is to find ions that can retain the unique photoelectric properties of lead halide perovskite and have low or even non-toxic materials to replace lead. In recent years, researchers have become interested in lead-free perovskite solar cells, which, although not as efficient as lead-based perovskite solar cells at this stage, show promising prospects.

Because lead halide perovskite material has excellent photoelectric performance, it is not only difficult to find substitutes for lead with low toxicity and easy processing properties, but also it requires direct band gap absorption of strong light and photon cycling to show excellent photovoltaic properties [67]. In fact, the promising alternatives to Pb are Sn/Ge halides, some double perovskites, and some Bi/Sb halides with perovskite-like structures (Figure 8). Specifically speaking, tin is the most promising element to replace lead. Tin halide perovskite has similar crystal structure to lead perovskite with $ABX_3$ lattice [68,69]. Sn-based perovskite has a narrower optical band gap and a higher carrier mobility, so it should theoretically have better optical properties [69,70]. Tin-based perovskites are represented by methylammonium tin oxide ($MASnI_3$), formamidine tin iodide ($FASnI_3$) and cesium tin iodide ($CsSnI_3$), which have direct band gaps of about 1.20, 1.41, and 1.3 eV, and are narrower than that of lead-based perovskites [71]. Sn has outer lone pair electrons. Theoretically, all Pb in $MAPbI_3$ can be replaced by Sn to form $MASnI_3$, which has a smaller band gap and a larger absorption coefficient than $MAPbI_3$. In theory, it should have better photovoltaic performance. All-halide $MASnX_3$ perovskite films, which can be processed from solution and exhibit good crystal quality, are expected to compete with existing photovoltaic technologies [72,73]. Unfortunately, compared with the heavier Pb elements in the 14th group of the periodic table, Sn elements with an electronic structure of $ns^2np^2$ have weaker interactions and are easily oxidized from $Sn^{2+}$ to $Sn^{4+}$ [68,74]. Therefore, tin-based perovskite solar cells are usually treated in glove boxes to prevent rapid degradation in the air. The researchers found that this effect could be suppressed by adding a reducing agent. For example, Hoshi et al. introduced $HOOC(CH_2)_4NH_3I$ (5-AVAI) into $MASnI_3$ to alleviate the oxidation of $Sn^{2+}$, which significantly improved the stability of the device [75]. $FASnI_3$ is another tin-based perovskite with excellent properties. The cationic radius of FA is slightly larger than that of MA. Replacing MA with FA can effectively reduce the oxidation degree from $Sn^{2+}$ to $Sn^{4+}$, making $FASnI_3$ more stable than $MASnI_3$ at room temperature [76]. The crystallization rate of $FASnI_3$ is much faster, resulting in abundant trap states and lower open circuit voltage ($V_{OC}$). To resolve this problem, Meng Xiang

Yue et al. introduced hydrogen bonds into FASnI$_3$ by adding polyvinyl alcohol (PVA) [77]. These hydrogen bonds introduced nucleation sites, which could slow the growth of crystals, guide crystal orientation, and reduce trap states. The inhibition of iodide migration greatly improved the stability of FASnI$_3$, with the conversion efficiency reaching 8.9%. CsSnX$_3$ is an all-inorganic Sn-based perovskite, where X can be I or Br. The black phase of CsSnX$_3$ is a direct bandgap semiconductor, with a hole mobility of $\mu_h \approx 585$ cm$^2$ V$^{-1}$ s$^{-1}$ and a carrier concentration of $\approx 10^{17}$ cm$^{-3}$, showing good photoelectric performance [78]. Maning Liu et al. found that Ge$^{2+}$ could not only stabilize Sn$^{2+}$ cations, but also enhance their optical and physical properties. They effectively filled the high-density ground Sn vacancy, reduced surface defects and increased photoluminescence quantum yield by partially replacing Sn atoms in nanostructures with Ge atoms [79]. As an all-inorganic perovskite, CsSnX$_3$ has better thermal stability than traditional organic-inorganic perovskite. As a less toxic element, Sn$^{2+}$ degrades to environmentally sound tin oxide when exposed to air, which is more beneficial to the environment than lead-based perovskite solar cells [80].

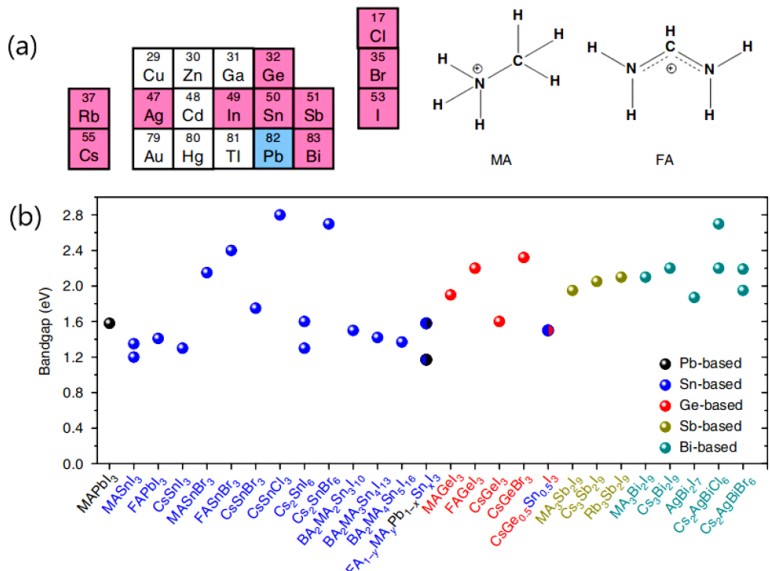

**Figure 8.** Potential solar cell absorption materials and band gaps of various materials. Reproduced with permission from [69]. Copyright 2019, Springer Nature.

Germanium, which belongs to the same main group as lead, is another element of interest to scientists. However, considering the small ionic radius of Germanium, poor solubility in polar solvents, and relatively wide band gap (1.6 eV), the PSC development rate of Germanium is much lower than that of tin PSC, Germanium PCE accounts for less than 5% [80–82].

Lead-free double perovskite has attracted much attention as a promising environmentally friendly photovoltaic material due to its inherent thermodynamic stability, appropriate band gap, small carrier effective mass, and low exciton binding energy [83]. An innovative strategy is to replace two Pb$^{2+}$ ions with a univalent cation and a trivalent cation to form a chartable double perovskite in the shape of A$_2$B$^I$B$^{III}$X$_6$ [84]. Cuncun Wu et al. successfully produced high-quality, highly stable double perovskite Cs$_2$AgBiBr$_6$ thin films using the low-pressure assisted (LPA) method (Figure 9a,b), and employed the films to produce planar heterojunction solar cells with an efficiency of 1.44% [85]. Scanning electron microscopy (SEM) images and film photographs (Figure 9c,d) showed that the LPA film presented a dense and smooth state. It was exciting that Cs$_2$AgBiBr$_6$ films exhibited good moisture, photostability, and thermal stability. The crystals lasted 240 days in the ambient atmosphere, showing no signs of decomposition, and even the chemical bonds remained the same. Besides, double perovskites composed of Cs$_2$AgInCl$_6$ are usually doped with various elements and have attracted attention for their superior optical properties, namely

self-trapping exciton (STE) emission and dopant induced photoluminescence. By alloying or doping in $Cs_2AgInCl_6$, it is possible to break the dual forbidden transition, change the band gap, and ultimately enhance the optical emission characteristics. Zhiguo Xia et al. studied $Cs_2AgInCl_6$ and proved that $Cs_2AgInCl_6$ had a direct band gap and a long carrier life and could be easily handled. The team demonstrated that $Cs_2AgInCl_6$ perovskite was thermodynamically stable and had several advantages over organic-inorganic perovskite [83].

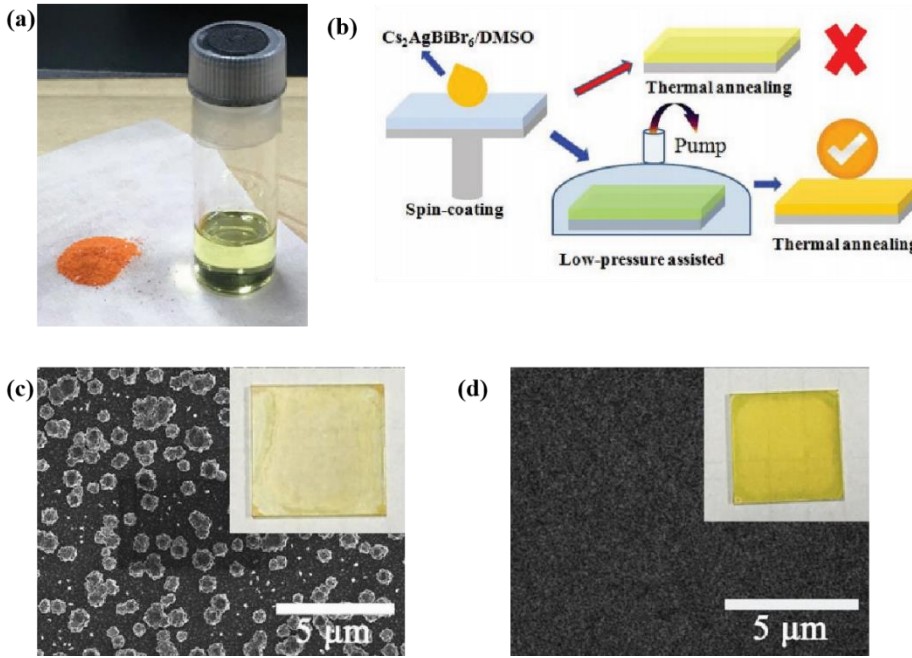

**Figure 9.** Preparation and SEM image of $Cs_2AgBiBr_6$ thin films. (**a**) Image of $Cs_2AgBiBr_6$ powder (left) and DMSO solution (right). (**b**) Film production process diagram. (**c,d**) SEM images of thin films obtained by (**c**) TA and (**d**) LPA. Reproduced with permission from [85]. Copyright 2018, Advanced science.

Lead-free inorganic copper-silver-bismuth halide materials may become a new development direction in the field of lead-free perovskite because of their environmental friendliness, high element abundance and low cost. Recently, Erchuang Fan et al. prepared inorganic lead-free $Cu_aAg_{m1}Bi_{m2}I_n$ absorption layers with a direct band gap of 1.78 eV by using low temperature gas-solid phase dispersion induced direct metal Surface element reaction (DMSER). At the same time, $Cu_aAg_{m1}Bi_{m2}I_n$/CuI bilayer films were prepared by one step low temperature gas-solid phase dispersion induced elemental reaction. The team through the FTO/TiO2/$Cu_aAg_{m1}Bi_{m2}I_n$/CuI/carbon structure solar cells, and obtained the PCE of 2.76%, for high efficiency, environmental protection provides a possible way to photovoltaic field [86].

## 6. Conclusions and Outlook

It remains a problem as to how to prevent the leakage of lead and effectively reduce the harm to the environment in order to achieve the wide commercial application of PSCs. Although the content of lead in PSCs is highly small, it is far from negligible in the current context. In general, the main methods to solve lead leakage at this stage are packaging, chemical adsorption, and lead-free PSCs. Firstly, packaging is the most common method to protect devices, which can greatly improve the strength of devices, powerfully resist the impact of the external environment, and effectively prevent lead leakage and improve the stability of devices. However, an important issue faced by packaging method is the aging of devices. Antioxidants need to be added to improve the anti-aging and stability

of packaging materials. At the same time, if the PV module is damaged and cannot be recovered for a long time, some lead will leak into the environment. Secondly, compared with the encapsulation method, chemical adsorption has such characteristics as low cost and high absorption efficiency. Chemisorption reduces the amount of lead leakage by 99.9% in certain cases through a series of chemical reactions during the lead leakage process to adsorb the lead in the device, or to adsorb the lead in the water. In the meantime, chemisorption can also be combined with physical packaging to achieve the lowest amount of lead leakage and reduce the pollution to the water source and the environment. Though chemisorption method has a good application prospect, further research is in need to achieve its commercial application. Thirdly, lead-free PSCs has developed rapidly in the past few years. At present, tin-based perovskite solar cells are the most competitive products to replace lead-based perovskite. Meanwhile, Germanium-based perovskite and double perovskite are developing stably. The biggest challenges confronting these lead-free perovskite products are the issues of product stability, conversion efficiency, and cost, which should be further explored by researchers. If these problems are solved, lead-free perovskite solar cells will be commercialized as low-toxicity or even non-toxic perovskite solar cells. We believe that PSCs can be made safer and more reliable through a combination of physical packaging and chemisorption, and by finding a strategy to recover lead from the environment. The search should continue for new lead-free, non-toxic materials to address concerns about the safety of lead.

**Author Contributions:** K.M. contributed to methodology, investigation, writing—original draft preparation; X.L. contributed to validation, data curation, formal analysis; F.Y. contributed to data curation, formal analysis, writing—review and editing; H.L. contributed to writing—review and editing, supervision. All authors have read and agreed to the published version of the manuscript.

**Funding:** This work was supported by the Henan Province college youth backbone teacher project (No.2020GGJS062).

**Institutional Review Board Statement:** Not applicable.

**Informed Consent Statement:** Not applicable.

**Data Availability Statement:** Data will be made available on request.

**Conflicts of Interest:** The authors declare that they had no known competing financial interests or personal relationships that could had appeared to influence the work reported in this paper.

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
