# Peer review of "Lead Leakage of Pb-Based Perovskite Solar Cells"

_coatings, doi:10.3390/coatings13061009_

Round 1

Reviewer 1 Report

The manuscript ‘Lead Leakage of Pb-based Perovskite Solar Cells’ authored by Ma et al. reviewed the main methods to reduce the lead leakage of Pb-based perovskite solar cells from the aspects of encapsulation, chemical adsorption and lead-free perovskite solar cells and then provided some prospect for further developing lead-free devices. The topic of lead leakage is important in the application of perovskite solar cells towards their practical use. However, I believe a major revision is required before further considering the acceptance of this manuscript.

A couple of comments:

1.    The novelty of this review is not clear or has not been highlighted. There are quite many good reviews that concern the toxicity and leakage issues of Pb in the application of perovskite solar cells. The authors should clearly state the significance of this review compared to others.

2.     There are only 7 references in the Introduction section, which number is too low. The authors should cite more recent related reports.

3.    In the Abstract: PCE should be power conversion efficiency and Line 13 it is weird to say ‘improve the problem of Pb leakage’. Please rephrase.

4.    Line 33, ‘ammonium formamide’ is not a typical A site cation, I believe the authors tended to say ‘methylammonium’. Please clarify.

5.     Line 34, rephrase the sentence ‘Although…’

6.    Line 111, not full name but only surname like ‘Jiang et al.’. Please check all cases and rectify.

7.     The authors should add one summarized table to compare the advantages and disadvantages of different methods, i.e., encapsulation and chemical adsorption.

8.   There are some issues for Figures. For example, it seems there are some watermarks in the background of Figures 4a and 4b. Please make sure to use the original high-resolution figures with the reproduction permission. Similarly, Figure 5 and Figure 7(c-e) are a bit vague.

9.     Section 5 should be ‘Conclusions and outlook’. The authors list some challenges but have not proposed possible solutions, which is important to provide hints for further development.

10.  In the section of Lead-free perovskite solar cells, some recent advanced reports should be properly cited, e.g., https://doi.org/10.1002/anie.202008724, https://doi.org/10.1002/solr.202100034, https://doi.org/10.1021/acsenergylett.3c00282.

Moderate editing of English language is required.

Author Response

Thank you for offering us an opportunity to improve the quality of our submitted manuscript (Lead Leakage of Pb-based Perovskite solar cells). We appreciated very much for the reviewer's constructive and insightful comments. In this revision, we have addressed all of these comments/suggestions. We hope the revised manuscript has now met the publication standard of your journal.

Comment 1 “The novelty of this review is not clear or has not been highlighted. There are quite many good reviews that concern the toxicity and leakage issues of Pb in the application of perovskite solar cells. The authors should clearly state the significance of this review compared to others.”

Reply: Thanks for this comment. Pb-based perovskite solar cell experienced a very rapid increase in both power conversion efficiencies (PCEs) and long-term stability in the past years. However, Pb toxicity is an inevitably problem. In this manuscript, we summary some relatively simple and practical approach about lead leakage suppression. It might provide new ideas for the further optimize perovskite solar cells. Addition, we have revised the abstract and clear the significance of this review.

Comment 2 “There are only 7 references in the Introduction section, which number is too low. The authors should cite more recent related reports.”

Reply: Thanks for your suggestion. We have added some latest studies in our manuscript.

Comment 3 “In the Abstract: PCE should be power conversion efficiency and Line 13 it is weird to say ‘improve the problem of Pb leakage’. Please rephrase.”

Reply: Thank you for this comment. We have checked the logic of the abstract and revised it carefully.

Comment 4 “Line 33, ‘ammonium formamide’ is not a typical A site cation, I believe the authors tended to say ‘methylammonium’. Please clarify.”

Reply: I’m sorry for this mistake. We have replaced the site cation from ‘ammonium formamide’ by ‘methylammonium’.

Comment 5 “Line 34, rephrase the sentence ‘Although…’”

Reply: We have corrected it to ‘Although the photovoltaic properties of pb-based perovskite solar modules are excellent, the safety hazard brought by them cannot be ignored’.

Comment 6 “Line 111, not full name but only surname like ‘Jiang et al.’ Please check all cases and rectify.”

Reply: This comment is absolutely right. We checked and rectified all the errors in our manuscript.

Comment 7 “The authors should add one summarized table to compare the advantages and disadvantages of different methods, i.e., encapsulation and chemical adsorption.”

Reply: Thanks for your constructive advice. We have added a comparative diagram of the advantages and disadvantages of encapsulation and chemisorption in Table 1.

Comment 8 “There are some issues for Figures. For example, it seems there are some watermarks in the background of Figures 4a and 4b. Please make sure to use the original high-resolution figures with the reproduction permission. Similarly, Figure 5 and Figure 7(c-e) are a bit vague.”

Reply: Thank you for your warning. We have obtained permission from the publisher to reproduce and adjust the sharpness of the images.

Comment 9 “Section 5 should be ‘Conclusions and outlook’. The authors list some challenges but have not proposed possible solutions, which is important to provide hints for further development.”

Reply: This is a good suggestion and we accepted it. We have accepted this suggestion and proposed some possible solutions.

Comment 10 “In the section of Lead-free perovskite solar cells, some recent advanced reports should be properly cited, e.g”

Reply: Thank you for the suggestion. We've added some recent advanced research to the lead-free section.

Thank you and best regards.

Sincerely yours,

Pro. Hairui Liu

29-04-2023

Reviewer 2 Report

The authors present the current state of knowledge on the reduction of lead leakage from perovskite PV modules to the environment: method of encapsulation using EVA foil: the method of chemical adsorption and the technology of lead-free cells.

However, all the figures and descriptions under the figures are copies from other articles. In the descriptions under the figures there are literature references but for two figures these are not source articles (fig. 1 and 9). There is also no permission from publishers for reproductions.

There are a lot of errors in the article and the language is unacceptable.

My comments:

1. All the figures and descriptions under the figures are copies from other articles. In the descriptions under the figures there are literature references but for 2 figures these are not source articles (fig. 1 and 9). There is also no permission from publishers for reproductions.

2. There are no references to figures in the text (except Fig.3 and Fig.4)

3. There are a lot of errors in the article and the language is unacceptable.

78. equipment – should solar cel or PV module

92. EVA wafer  - should be EVA sheet

99. EVA tablets

117. 21pb    

 Many unclear or incorrect sentences:

209.Various defects in perovskite films can form deep energy wells….

157-159. flexible perovskites usually need to be encapsulated in organic materials because organic packaging materials can be synthesized through organic molecules with specific properties by changing the energy level, molecular weight and solubility of their own molecules [19].

247. stability of the period

262. 23% with negligible latency

280. Due to the excellent photovoltaic properties of lead,

282. but also it requires direct bandgap absorption of strong light and photon cycling to show excellent photovoltaic

288, ammonium tin iodide (should be methylammonium tin oxide)

315 The orthogonal black phase of CsSnX3

317 ref [39] applies to MAPbI3 and not CsSnX3

Author Response

Thank you for offering us an opportunity to improve the quality of our submitted manuscript (Lead Leakage of Pb-based Perovskite solar cells). We appreciated very much for the reviewer's constructive and insightful comments. In this revision, we have addressed all of these comments/suggestions. We hope the revised manuscript has now met the publication standard of your journal.

Comment 1 “equipment – should solar cel or PV module”

Reply: Thanks for pointing out the mistake. We have revised it in the manuscript.

Comment 2 “EVA wafer  - should be EVA sheet” “EVA tablets”

Reply: We completely accept this comment and have revised it in the manuscript.

Comment 3 “21pb”

Reply: We are sorry for this typing-mistake. We have revised it in the manuscript.

Comment 4 “23% with negligible latency”

Reply: We apologize for this mistake. It should be ‘hysteresis’ but not ‘latent’. The manuscript has been checked and revised carefully.

Comment 5 “Due to the excellent photovoltaic properties of lead”

Reply: We are sorry for this ambiguous expression. We have replaced that sentence by ‘the Pb-based halide perovskite material has excellent photoelectric performance’.

Comment 6 “All the figures and descriptions under the figures are copies from other articles. In the descriptions under the figures there are literature references but for 2 figures these are not source articles (fig. 1 and 9). There is also no permission from publishers for reproductions”

Reply: This is a terrible mistake. Thank you for your reminding. We have obtained the publisher's permission to reproduce and changed the description under the image.

Comment 7 “There are a lot of errors in the article and the language is unacceptable.”

Reply: Thank you very much for this comment. We have polished our manuscript by professional bodies.

Comment 8 “There are no references to figures in the text (except Fig.3 and Fig.4)”

Reply: Thanks for your suggestion. We've checked our manuscript carefully and added references to all the images.

Thank you and best regards.

Sincerely yours,

Pro. Hairui Liu

29-04-2023

Reviewer 3 Report

The manuscript submitted by Liu and co-workers aims at reviewing the issue of Pb leakage from Perovskite Solar Cells. This is a rising topic in PSCs field, no matter the relatively low amount of Lead in this class of emerging photovoltaic. Unfortunately, the review fails in pursuing this aim. Indeed, the main cause in Pb leakage is relating to operational flaws and, as such, the encapsulation section (first method) should be extend and different class of encapsulant materials (such as ALD layer or thermosetting polymers) should be discussed. Moreover, the authors just reported some literature findings but the discussion of the latter is very poor: indeed, it is not clear how the different encapsulation strategies helped in reducing Pb leakage. 

Similarly, in the second section the addition of protection layers should be better discussed in relation with the final photovoltaic efficiency and some strategy to reach the best tradeoff between efficiency and low leakage should be discussed/proposed. 

A serious flaw of this manuscript is the absence of a section in which the mechanism for Pb leakage and its dispersion in the environment are discussed. As a matter of fact, Pb leakage in PSCs is a more critical issue compared to other technologies because here Pb is ususally bound to ammonium cation. The latter plays a dramatic role in promoting the leakage of PB by the formation of mixed complexes that could more easily adsorbed by plants and microorgnisms. Authors should better consider this aspects. For example the use of some adsorption systems (second section) could be an added value to prevent the leakage during the operative lifetime of the device but could be a drawback promoting Lead or Lead/ammonium systems dispersion in water. 

Last section, dealing with Pb-replacement in PSCs, should be deleated. From one side, it is obvious that the absence of Lead will results in zero leakage (but, as such, one could ask why other technologies are not discussed though); from the other, the main alternative to Pb-PSCs are Sn-PSCs, but Tin has its own concerns from a sustainability and environmental-friendness point of view and  thus it is not a valid alternative to Pb, also considering that its bio-accumalation is much more easier compared to Pb ones. 

All in all, the idea behind this manuscript is of interest and timing. Yet, it should be better organized and discussed. Albeit this reviewer will be glad to review a deeply revised version of this manuscript, the latter could not be accepted in the present form. Revision required are too extensive to allow a "major revision" decision as well.

English is fine.

Author Response

Thank you for offering us an opportunity to improve the quality of our submitted manuscript (Lead Leakage of Pb-based Perovskite solar cells). We appreciated very much for the reviewer's constructive and insightful comments. In this revision, we have addressed all of these comments/suggestions. We hope the revised manuscript has now met the publication standard of your journal.

Comment 1: “A serious flaw of this manuscript is the absence of a section in which the mechanism for Pb leakage and its dispersion in the environment are discussed.”

Reply: Thank you for this good suggestion. We have added a section devoted to make clear the mechanism of lead leakage. The mechanism of lead leakage under different conditions was discussed in this section.

Comment 2: “Last section, dealing with Pb-replacement in PSCs, should be deleated.”

Reply: Thank you for your suggestion. We think that preparing the low lead, even lead-free, perovskite is also one of methods to solve the problem of lead leaks. And an increasing effort is devoted to mitigating the Pb issue by searching for Pb-free halide perovskite alternatives. Therefore, we think the section about Pb-replacement in PSCs is needed.

Comment 3: “the encapsulation section (first method) should be extend and different class of encapsulant materials (such as ALD layer or thermosetting polymers) should be discussed.”

Reply: The reviewer is right. We've added a discussion about encapsulant materials and how can reduce lead leaks. The encapsulation layer can greatly improve the impact strength of the device, and can effectively block the perovskite decomposition caused by water and oxygen penetration to prevent lead leakage.

Comment 4: “some strategy to reach the best tradeoff between efficiency and low leakage should be discussed/proposed.”

Reply: Thanks to the reviewer for this comment. It is a tricky issue that the balance between efficiency and low leakage. Many researchers develop to achieve high efficiency perovskite solar cells with low lead leakage. We have added some discussion about it in the manuscript.

Thank you and best regards.

Sincerely yours,

Pro. Hairui Liu

29-04-2023

Reviewer 4 Report

The review paper deals with one of the important challenge perovskite commercialization is facing. While it describes the main methods (encapsulation, chemical adsorption and lead-free perovskites) that have been applied for mitigating the lead leakage issue, an addition of more reports is suggested to better capture the progress achieved so far. For example there is an extended description of the Cation exchange resin, nevertheless, a series of other molecules have been incorporated into the perovksite that can effectively trap the Pb+2 though chemical absorption. Thus, an update of the review with a some more recent published work, especially in the chemical adsorption, is recommended.

A Moderate editing of English language is suggested, for example:

- some definitions such as "equipment" when referring to devices or modules( eg. line 248) must be reconsidered,

- missing verb in line 238, 

- line 212 needs to be rephrased,

- and a statement seems to exaggerate in line 258 stating that "Junjun Chen solved the leak problem".

Author Response

Thank you for offering us an opportunity to improve the quality of our submitted manuscript (Lead Leakage of Pb-based Perovskite solar cells). We appreciated very much for the reviewer's constructive and insightful comments. In this revision, we have addressed all of these comments/suggestions. We hope the revised manuscript has now met the publication standard of your journal.

Comment 1: “some definitions such as "equipment" when referring to devices or modules( eg. line 248) must be reconsidered”

Reply: The reviewer is absolutely right here. All the proper nouns in the manuscript have been redefined, such as ‘PV module’.

Comment 2: “line 212 needs to be rephrased”

Reply: Thanks to the reviewer for this comment. We have revised the paragraph that coating CER on the glass surface can form a good covering layer, so that the glass surface has stronger hydrophilicity.

Comment 3: “and a statement seems to exaggerate in line 258 stating that "Junjun Chen solved the leak problem"”

Reply: It is a poor expression. We have replaced it by ‘provided a new way to solve the problem of lead leakage’.

Thank you and best regards.

Sincerely yours,

Pro. Hairui Liu

29-04-2023

Round 2

Reviewer 1 Report

The authors have addressed most of my previous comments. However, there are still two concerns which should be carefully addressed before the acceptance.

1. There are still quite number of typos throughout the manuscript, which require the authors to double check and correct, e.g., page 8 atomic Layer deposition, page 10 Figure 5 caption MAPbI3 (subscript, check all other cases), page 17 the structure FTO/TiO2 /CuAgBi2I8 CuaAgm1Bim2In CuI/carbon' the active layer expression looks a bit weird, CuI is a typical hole-transport material, please clarify.

2. The suggested references (https://doi.org/10.1002/anie.202008724, https://doi.org/10.1002/solr.202100034, https://doi.org/10.1021/acsenergylett.3c00282) have not been cited properly in the section of Lead-free perovskite solar cells. They are representative studies on the development of Sn-based perovskite solar cells both in the form of thin film and nanocrystals. 

There are still quite a few typos throughout the manuscript, which require the authors to double check and correct

Reviewer 2 Report

Most of my comments have been taken into account, but I have a few more comments.

1. There are no references to figures in the text.

2. Copyright for Fig. 1, Fig. 8 and Fig. 9 is missing.

3. Spelling mistakes:    - µh≈585 cm2V-1s-1 and a carrier concentration of ≈1017    - CuaAgm1Bim2In - FTO/TiO2 /CuAgBi2I8 CuaAgm1Bim2In CuI/carbon structure solar

4. In the "Mechanism of lead leakage from perovskite", the authors write about the perovskite CH3NH3PbI3. Currently, however, mixed perovskites and FAPbI3 perovskite in combination with a 2D perovskite layer are more prospective because they are more stable.  

The English language still needs to be improved.

Reviewer 3 Report

Author made some (small) effort to improve the overall quality of their mansucript, but the final results is not satisfactory. Indeed, they only add some small paragraphs to accomplish reviewers' request but the organization of the review still presents some major issues. 

1) Authors add just a couple of reference more, but this was not the aim of the comment. Indeed, considering operational failures as the main cause of Pb-leakage, the role of the encapsulant deserves a main role. EVA and Epoxy-resins are just a couple of polymers that could be used, moreover EVA itself it is quite unstable and could favour the cell failures. This section definetively deserves more attention (and text). 

2) Startegies to find out a trade-off between no-leakage and efficiency are only cited, but they are not critically analyzed or compared each others. A critical analyses of literature data is what one can expect from a review.

3) A (very) short section regarding the mechanism of lead leakage has been added, but the description of the possible mechanisms is quite poor. 

4) With respect to Pb-replacement section (number 5 in the revised version), this reviewer agrees that different groups all around the world are indefatigably working to replace Lead, yet, if this section should be added, the title should be revised: as a matter of fact, Lead leakage could be considered only if Lead is there. 

All in all, the changes made by the authors are only limited and the manuscript is not a the required level to be published.

see above

Reviewer 4 Report

I would like to thank the authors for the reply. All my comments were addressed, and I would suggest the publication of the paper in the present form.
